# Emergent Communication:
# Generalization and Overfitting in Lewis Games

**Mathieu Rita**
INRIA, Paris
mathieu.rita@inria.fr

**Corentin Tallec**   **Paul Michel**[*]   **Jean-Bastien Grill**
DeepMind
[corentint,paulmiche,jbgrill]@deepmind.com

**Olivier Pietquin**
Google Research, Brain Team
pietquin@google.com

**Emmanuel Dupoux**
EHESS,ENS-PSL,CNRS,INRIA
Meta AI Research
emmanuel.dupoux@gmail.com

**Florian Strub**
DeepMind
fstrub@deepmind.com

## Abstract

Lewis signaling games are a class of simple communication games for simulating the emergence of language. In these games, two agents must agree on a communication protocol in order to solve a cooperative task. Previous work has shown that agents trained to play this game with reinforcement learning tend to develop languages that display undesirable properties from a linguistic point of view (lack of generalization, lack of compositionality, etc). In this paper, we aim to provide better understanding of this phenomenon by analytically studying the learning problem in Lewis games. As a core contribution, we demonstrate that the standard objective in Lewis games can be decomposed in two components: a co-adaptation loss and an information loss. This decomposition enables us to surface two potential sources of overfitting, which we show may undermine the emergence of a structured communication protocol. In particular, when we control for overfitting on the co-adaptation loss, we recover desired properties in the emergent languages: they are more compositional and generalize better.

## 1   Introduction

Understanding the dynamics of language evolution has been a challenging if not controversial research topic in the language sciences [32, 13]. Given that the very first human language cannot be unearthed from fossils [5], computational models have been designed to simulate the emergence of a structured language within a controlled environment. In this line of work, Lewis signaling games [55] are among the most widespread playground environments to model language emergence: they are inherently simple, yet they exhibit a rich set of communication behaviors [17, 70]. Therefore, understanding Lewis games dynamics may shed light on the prerequisites of language emergence.

In their original form, Lewis signaling games involve two agents: a speaker and a listener. The speaker observes a random state from its environment, e.g. an image, and sends a signal to the listener. The listener then undertakes an action based on this signal. Finally, both agents are equally rewarded based on the outcome of the listener's action. The resolution of this cooperative two-player game requires the emergence of a shared protocol between the agents [55, 17]. One way to model the emergence of such protocol is to give the agents the capacity to learn. The agents, and therefore, the communication protocol, are shaped by a sequence of trials and errors over multiple games [81, 44, 75, 70]. This learning-centric approach allows for a fine analysis of the language emergence dynamics [70, 36]. It

---

[*]This work was performed when Paul Michel was affiliated with Ecole Normale Supérieure PSL.

36th Conference on Neural Information Processing Systems (NeurIPS 2022).

also raises challenging learning-specific questions: What are the inductive biases present in the agent architecture and loss function that shape the emergent language [43]? How do agents generalize from their training set? Is the resulting language compositional [8]? What is the impact of overfitting [53]?

Recently, there has been a resurgence of interest for such learning-based approaches following advances in machine learning [51]. In these approaches, the speakers and listeners are modeled as deep reinforcement learning agents optimized to solve instances of the Lewis games [53, 33, 65, 56, 28]. The vast majority of these works explore Lewis games from an empirical perspective. However, some of the recent experimental results are at odds with experimental findings from the linguistics literature. For instance, the emergent protocols lack interpretability [48], generalization does not always correlate with language compositionality [10], successful strategies are not naturally adopted in populations [68, 12], and anti-efficient communication may even emerge [9]. It is unclear whether those empirical observations result from a learning failure, e.g. optimization problems, overfitting, or whether they are symptomatic of more fundamental limitations of Lewis games for modeling language emergence, e.g. lack of embodiment [31, 4, 63, 37]. Overall, it is crucial to establish new analytical insight to analyze Lewis games in the learning setting.

In this paper, we introduce such an analytical framework to diagnose the learning dynamics of deep reinforcement learning agents in Lewis signaling games. As a core contribution, we demonstrate under mild assumptions that the loss of the speaker and listener can be decomposed into two components when resolving Lewis signaling games: (i) an *information loss* that maximizes the mutual information between the observed states and speaker messages; (ii) a *co-adaptation loss* that aligns the speaker and listener's interpretation of the messages (Section 2). Based on this decomposition, we empirically examine the evolution of these two losses during the learning process (Section 5). In particular, we identify an overfitting problem in the co-adaptation loss between the agents which undermines the emergence of structured language. We then show that the standard setup used in the deep language emergence literature consistently suffers from this overfitting issue (Section 5.1). This realization explains some of the contradictory observations [10] and experimental choices from past works [65, 56, 68]. Finally, we explore regularization methods to tackle this co-adaptation overfitting. We observe that reducing the co-adaptation overfitting allows for developing a more structured communication protocol (Section 5.2).

All in all, our contributions are three-fold: (i) we provide a formal description of Lewis games from a learning standpoint (Section 2.3); (ii) we apply this framework in experiments to show that degenerate results are primarily due to overfitting in the co-adaptation component of the game (Section 5.1) ; (iii) we propose natural ways of tackling this overfitting issue and show that, when we control the receiver's level of convergence, we obtain a well-structured emergent protocol (Section 5.2).

## 2 Analyzing Lewis Games

We show that Lewis games' objective decomposes into two terms: (i) an information loss that measures whether each message refers to a unique input; (ii) a co-adaptation loss that quantifies the alignment of the speaker's and listener's interpretation of the messages.

For simplicity and to ease the reader's intuition, we focus on the reconstruction variant of Lewis games with agents optimizing the reconstruction log-likelihood in the main paper. In Appendix A, we show that our analysis extends to a broader of Lewis signaling games, e.g. discrimination games [12, 60, 21, 30, 65, 53, 52, 33, 56, 58], and to a general form of reward that covers the rewards commonly used in emergent communication, e.g. log-likelihood [9, 10, 40, 68, 67, 11], accuracy reward [65, 53, 52, 48, 56, 28, 23].

### 2.1 Background: Lewis Reconstruction Games

**Game formalism** In reconstruction Lewis games, a speaker observes a random object of its environment. The speaker then sends a descriptive message, which a second agent, the listener, uses to reconstruct the object. The success of the game is quantified by how well the original object is reconstructed [40, 9, 67, 68]. Formally, the speaker is parameterized by $\theta$ and the listener is parameterized by $\phi$. The observed object denoted by $x$ is selected from a set of objects denoted by $\mathcal{X}$. We denote by $X$ the random variable characterizing $x$, sampled from distribution $p$. The intermediate message sent by the speaker $m$ belongs to the set of all potential messages $\mathcal{M}$. The speaker follows

a policy $\pi_\theta$ which samples a message $m$ with probability $\pi_\theta(m|x)$ conditioned on object $x$. We denote by $M_\theta$ the random variable characterizing the message $m$, sampled from $\pi_\theta(\cdot|X)$. We denote by $\pi_\theta(m) = \sum_x \pi_\theta(m|x)p(x)$ the marginal probability of a message given policy $\pi_\theta$. Given a message $m$, the listener outputs a probability distribution over inputs $\rho_\phi(\cdot|m)$, and the probability of reconstructing the entire object $x$ given $m$ is thus $\rho_\phi(x|m)$.

**Game objectives** In reconstruction games, the speaker and listener minimize the negative log likelihood of the reconstructed object. Both agents thus optimize the objective:

$$\mathcal{L}_{\theta,\phi} = -\mathbb{E}_{x\sim p, m\sim\pi_\theta(\cdot|x)}[\log \rho_\phi(x|m)], \tag{1}$$

where optimizing the speaker is a reinforcement learning problem whose parameters $\theta$ are optimized using policy gradient [76] and optimizing the listener is a supervised learning problem whose parameters $\phi$ are optimized with gradient descent. In our theoretical analysis, we consider that agents are not regularized. In practice, regularizations, e.g. entropy regularization [59], may be added to the game objective but it does not alter our main conclusions.

## 2.2 Building Intuition on the Lewis Reconstruction Game Learning Dynamics

To get a better intuition of the dynamic of Lewis reconstruction games, we can analyze the form taken by the optimal listener, given speaker $\pi_\theta$. In what follows, we use sub-script $\theta$ to denote an explicit dependency of the policy on parameters $\theta$, e.g. a policy parameterized with a neural network. Conversely, the use of super-script $*(\theta)$ corresponds to an implicit dependency of the policy on parameters $\theta$. As shown in Appendix A.1.1, given a message $m$, the optimal listener's distribution $\rho^{*(\theta)}(\cdot|m)$ can be written in closed-form:

$$\rho^{*(\theta)}(x|m) := \frac{p(x)\pi_\theta(m|x)}{\sum_{x'\in\mathcal{X}} p(x')\pi_\theta(m|x')}. \tag{2}$$

Here, $\rho^{*(\theta)}$ does not depend on $\phi$, but implicitly depends on $\theta$, as it is the optimal listener *given a policy parameterized by* $\theta$. At each update, the listener $\rho_\phi$ gets closer to its optimum $\rho^{*(\theta)}(\cdot|m)$. If we suppose that the listener perfectly fits $\rho^{*(\theta)}(\cdot|m)$ at any moment, the loss becomes:

$$\mathcal{L}_{\theta,\phi} = -\mathbb{E}_{x\sim p, m\sim\pi_\theta(\cdot|x)}[\log \rho^{*(\theta)}(x|m)] = \mathcal{H}(X|M_\theta) = -I(X;M_\theta) + \mathcal{H}(X) \tag{3}$$

where $\mathcal{H}(X|M_\theta)$ is the conditional entropy of $X$ conditioned on $M_\theta$ and $I(X;M_\theta)$ is the mutual information between $X$ and $M_\theta$. Thus, if the listener is optimal at every point in time, the speaker's task merely becomes the construction of a message protocol that maximizes the mutual information between objects and messages, i.e. the construction of an unambiguous message protocol.

In practice, the listener never perfectly fits the optimum. In the following, we elucidate the effect of this gap between the listener and its optimum on the dynamics of the game.

## 2.3 Analytical Result: The Lewis Games Loss Decomposition

In cooperative Lewis games, the agents' loss can be decomposed into two terms:

$$\mathcal{L}_{\theta,\phi} = \mathcal{L}_{\text{info}} + \mathcal{L}_{\text{adapt}}, \tag{4}$$

- An **information term** $\mathcal{L}_{\text{info}}$ quantifies the degree of ambiguity of the language protocol. It is minimal when each message refers to a unique object;
- A **co-adaptation term** $\mathcal{L}_{\text{adapt}}$ quantifies the gap between the listener and its optimum: the speaker's posterior distribution. This co-adaptive term is optimized both by the speaker and the listener. When the listener is optimal, this co-adaptation objective is zeroed.

In particular, the decomposition takes the following form in the Lewis reconstruction game:

$$\mathcal{L}_{\theta,\phi} = \underbrace{\mathcal{H}(X|M_\theta)}_{\mathcal{L}_{\text{info}}} + \underbrace{\mathbb{E}_{m\sim\pi_\theta} D_{KL}(\rho^{*(\theta)}(\cdot|m)||\rho_\phi(\cdot|m))}_{\mathcal{L}_{\text{adapt}}}, \tag{5}$$

The proof of the decomposition is provided in Appendix A. Appendix A provides the proof for the reconstruction log-likelihood reward and extends to a broader class of Lewis signaling games, e.g. discrimination games, and general cooperative rewards covering usual emergent communication rewards, e.g. the accuracy reward. This decomposition gives us insights on the game dynamics and the constraints that shape languages in the game with neural agents:

**The information loss** $\mathcal{L}_{\text{info}}$ captures the speaker's intrinsic objective: to develop an unambiguous protocol. $\mathcal{L}_{\text{info}}$ is minimal, equals to 0, when the communication protocol is unambiguous, i.e. every message from the speaker's policy $\pi_\theta$ refers to a unique object. Conversely, $\mathcal{L}_{\text{info}}$ is maximal, equal to $\mathcal{H}(X)$, when the message protocol is fully ambiguous, and $X$ and $M_\theta$ are independent variables.

**The co-adaptation loss** $\mathcal{L}_{\text{adapt}}$ is specific to learning agents. This loss measures how far the listener $\rho_\phi$ is from its optimum $\rho^{*(\theta)}$. If $\mathcal{L}_{\text{adapt}} = 0$, the listener and its optimum coincide. $\mathcal{L}_{\text{adapt}}$ has the particularity to be optimized by the two agents. From the listener's side, it merely corresponds to the optimization of its supervised task. From the speaker's side, it brings out that the speaker must adapt its language to the listener in addition to build an unambiguous message protocol. In other words, the co-adaptation loss pushes the speaker to develop a language that can be easily recognized by listeners. This pressure diminishes as the listener approaches its optimum.

From a practical perspective, Equation (5) yields the following individual gradients:

$$\begin{cases} \nabla_\theta \mathcal{L}_\theta & = -\nabla_\theta I(X, M_\theta) + \nabla_\theta \mathbb{E}_{m \sim \pi_\theta} D_{KL}(\rho^{*(\theta)}(\cdot|m)||\rho_\phi(\cdot|m)) \\ \nabla_\phi \mathcal{L}_\phi & = \nabla_\phi \mathbb{E}_{m \sim \pi_\theta} D_{KL}(\rho^{*(\theta)}(\cdot|m)||\rho_\phi(\cdot|m)), \end{cases} \tag{6}$$

where the listener only receives gradients from the co-adaptation term, and the speaker receives gradients from both terms.

This loss decomposition also finds echoes in the cognitive science literature in the form of an expressivity vs. learnability trade-off [72]; see Section 6 for a detailed discussion.

## 2.4 Generalization Gaps in Lewis Reconstruction Games

We explore another facet of the loss decomposition that arises from learning. As agents are trained on partial views of their environment, it opens questions of overfitting and generalization to unseen objects. As is customary in machine learning, we consider agents trained on a fixed, finite sample from the data distribution: the training set. Let us denote by $p_{\text{train}}$ the *empirical* object distribution over the training set and $X^{\text{train}}$ an object sampled from $p_{\text{train}}$. Similarly let $M_\theta^{train}$ denote a message sampled from $\pi_\theta(.|X^{\text{train}})$, $\pi_\theta^{\text{train}}(m) = \sum_x \pi_\theta(m|x)p_{\text{train}}(x)$ the marginal probability of a message on the training set, and $\rho_{\text{train}}^{*(\theta)}(x|m) = \frac{p_{\text{train}}(x)\pi_\theta(m|x)}{\sum_{x \in \mathcal{X}} p_{\text{train}}(x)\pi_\theta(m|x)}$ the speaker's posterior distribution with respect to the prior distribution $p_{\text{train}}$. The training loss can be written as follow:

$$\begin{aligned} \mathcal{L}_{\theta,\phi}^{\text{train}} &= -\mathbb{E}_{x \sim p_{\text{train}}, m \sim \pi_\theta(\cdot|x)}[\log \rho_\phi(x|m)] \\ &= \underbrace{\mathcal{H}(X^{\text{train}}|M_\theta^{\text{train}})}_{\mathcal{L}_{\text{info}}^{\text{train}}} + \underbrace{\mathbb{E}_{m \sim \pi_\theta^{\text{train}}} D_{KL}(\rho_{\text{train}}^{*(\theta)}(\cdot|m)||\rho_\phi(\cdot|m))}_{\mathcal{L}_{\text{adapt}}^{\text{train}}} \cdot \end{aligned}$$

Decomposing the gap between $\mathcal{L}_{\theta,\phi}^{\text{train}}$ and $\mathcal{L}_{\theta,\phi}$ uncovers two sources of overfitting:

$$\mathcal{L}_{\theta,\phi}^{\text{train}} = \mathcal{L}_{\theta,\phi} + \underbrace{\mathcal{L}_{\text{info}}^{\text{train}} - \mathcal{L}_{\text{info}}}_{\text{information overfitting}} + \underbrace{\mathcal{L}_{\text{adapt}}^{\text{train}} - \mathcal{L}_{\text{adapt}}}_{\text{co-adaptation overfitting}} \cdot \tag{7}$$

Intuitively, *information overfitting* occurs when the speaker only develops an unambiguous language on the training set, but ambiguities remain on the total dataset. *Co-adaptation overfitting* occurs when the two agents agree on a common communication protocol on the training data, but not on all data.

## 3 Method

This section gathers the methodological tools required to empirically study the loss decomposition.

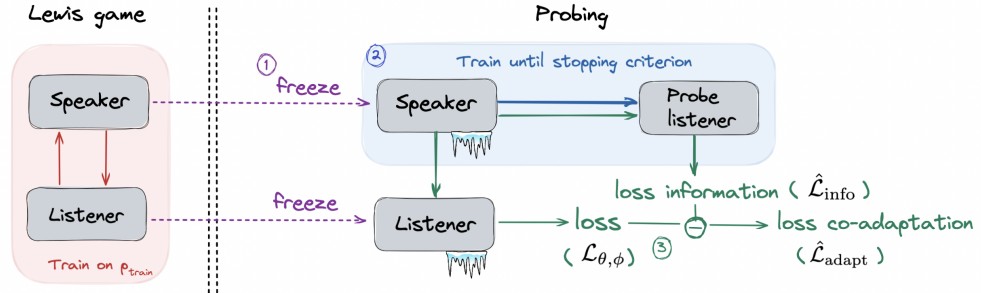

Figure 1: Probing method: (1) the speaker and listener are frozen and the probe listener is initialized. (2) the probe listener is trained on $p_{\text{train}}$ (resp. $p$) with the speaker's messages until convergence; (3) The speaker takes inputs from $p_{\text{train}}$ (resp. $p_{\text{test}}$) and messages the probe listener and the listener. The resulting loss of the probe listener is $\hat{\mathcal{L}}_{\text{info}}$, and the loss of the listener is used to estimate $\hat{\mathcal{L}}_{\text{adapt}}$.

### 3.1 Probing the Information and Co-adaptation Losses

Computing $\mathcal{L}_{\text{info}}$ and $\mathcal{L}_{\text{adapt}}$ directly necessitates estimating the posterior distribution of the speaker, $\rho^{*(\theta)}(.\mid m)$. Doing so requires summing over all $\mathcal{X}$ which is intractable. Fortunately, deep models are large enough so that they can perfectly solve their task on their train set. We can leverage this fact to compute empirical estimates $\hat{\mathcal{L}}_{\text{info}}$ and $\hat{\mathcal{L}}_{\text{adapt}}$ of $\mathcal{L}_{\text{info}}$ and $\mathcal{L}_{\text{adapt}}$ respectively by using an auxiliary listener trained to optimality.

We here detail an empirical probing mechanism to obtain estimates $\hat{\mathcal{L}}_{\text{info}}$ and $\hat{\mathcal{L}}_{\text{adapt}}$ given speaker $\pi_\theta$ and listener $\rho_\phi$. As noted in Equation 2, the posterior $\rho^{*(\theta)}$ also corresponds to the optimal listener. Therefore, we obtain an estimate of the posterior by training a listener to optimality, and use this optimal listener to decompose the loss. In practice, to obtain this optimal listener, we freeze speaker $\pi_\theta$ and listener $\rho_\phi$ and initialize a new, auxiliary listener from scratch, which we refer to as the *probe* listener. As illustrated in Figure 1, the probe listener is trained to reconstruct object $x$ from message $m$, with $x$ drawn from distribution $p$ or $p_{\text{train}}$ and $m$ sampled according to the frozen speaker policy $\pi_\theta(.|x)$, until a stopping criterion is met. We then distinguish between the train and test estimates:

$$\begin{aligned}
\hat{\mathcal{L}}_{\text{info}}^{\text{train}} &= -\mathbb{E}_{x \sim p_{\text{train}}, m \sim \pi_\theta(\cdot|x)}\left[\log \rho_{\omega^*}^{\text{train}}(x|m)\right] \\
\hat{\mathcal{L}}_{\text{adapt}}^{\text{train}} &= -\mathbb{E}_{x \sim p_{\text{train}}, m \sim \pi_\theta(\cdot|x)}\left[\log \rho_\phi(x|m)\right] - \hat{\mathcal{L}}_{\text{info}}^{\text{train}}
\end{aligned} \tag{8}$$

and,

$$\begin{aligned}
\hat{\mathcal{L}}_{\text{info}}^{\text{test}} &= -\mathbb{E}_{x \sim p_{\text{test}}, m \sim \pi_\theta(\cdot|x)}\left[\log \rho_{\omega^*}(x|m)\right] \\
\hat{\mathcal{L}}_{\text{adapt}}^{\text{test}} &= -\mathbb{E}_{x \sim p_{\text{test}}, m \sim \pi_\theta(\cdot|x)}\left[\log \rho_\phi(x|m)\right] - \hat{\mathcal{L}}_{\text{info}}^{\text{test}}
\end{aligned} \tag{9}$$

where $\rho_{\omega^*}^{\text{train}}$ and $\rho_{\omega^*}$ are the probe listeners trained over distributions $p_{\text{train}}$ and $p$ respectively.[2] Note that this probing mechanism, while tractable, is computationally costly as it necessitates training a new probe listener to convergence, and so we only use it as a valuable diagnosis tool.

### 3.2 Balancing the Information and Co-adaptation Terms

As explained in Section 2.3, the information loss alone is sufficient for the speaker to develop an unambiguous language. This begets the question: does the co-adaptation loss have any bearing on the emergent language at all? We elucidate this question by balancing the weight of the co-adaptation term in the decomposition. By using the probing method described above, we build the following training loss:

$$\mathcal{L}_\theta(\alpha) = (1-\alpha) \times \hat{\mathcal{L}}_{\text{info}}^{\text{train}} + \alpha \times \hat{\mathcal{L}}_{\text{adapt}}^{\text{train}} \qquad \text{where} \quad \alpha \in [0; 0.5] \tag{10}$$

---

[2]The estimate is trained on $p$, the *full* distribution of objects, and not $p_{\text{test}}$. Training on $p_{\text{test}}$ would result in an optimal listener overfitting on the test set, which would results in bad estimates of the mutual information.

Hence, $\alpha$ balances the two speaker objectives (up to an approximation error). When $\alpha = 0.5$, the loss falls back to the classic setting. When $\alpha = 0$, the co-adaptation term is removed on the speaker side; note that the Lewis game can still be solved since the listener still optimizes the co-adaptation term. We experimentally analyse the effect of $\alpha$ on resulting languages in Section 5.1. In Appendix B, we describe how we build the balanced loss and explain why $\alpha$ should be bounded by 0.5.

### 3.3 Controlling the Listener's Co-adaptation Loss Level of Convergence

As mentioned in 2.2, the influence of $\mathcal{L}_{\mathrm{adapt}}$ on the co-adaptation term in the speaker's loss is modulated by the listener's level of convergence to its optimum. To understand the effect of this co-adaptation, we decouple the speaker and listener training and train the listener via three procedures:

**Continuous listener** The listener is continuously trained, jointly with the speaker. This is the standard setting in the emergent communication literature, and serves to report the baseline behavior.

**Partial listener** The listener is re-initialized *after each* of the speaker's update and trained on the training set for $N_{step}$ before updating the speaker again. This baseline enables fine-grained analysis of the influence of under-training (low $N_{step}$) and over-training (large $N_{step}$) the listener.

**Early stopping listener** The listener is also re-initialized *after each* of the speaker's update but is now trained until an early stopping criterion is met on the validation set. This allows us to get the best estimate of the posterior $\rho^{*(\theta)}(.|m)$ at each update. This can be seen as a variant of the partial listener with an adaptive number of steps $N_{step}$.

## 4 Experimental settings

### 4.1 Game description

Unless specified, all our experiments are run on the reconstruction game defined in Section 2.1. Experiments are run over 6 seeds and reach $> 99\%$ training reconstruction scores unless otherwise stated. Our implementation is based on the EGG toolkit [39] and the code is available at `https://github.com/MathieuRita/Population`.

**Environment** We consider objects $x =: (x_1, ..., x_K) \in \mathcal{X} =: \mathcal{X}_1 \times ... \times \mathcal{X}_K$ characterized by $K$ attributes where attribute $i$ may take $|\mathcal{X}_i|$ different values. By design, this synthetic environment allows us to test the ability of agents to refer to unseen objects by communicating their attributes [3, 48]. Each object is the concatenation of one-hot representations of the attributes $(x_i)_{1 \leq i \leq K}$. Objects have $K = 6$ attributes, each taking 10 different values, for a total of 1 million objects. Training, validation and test sets are randomly drawn from this pool of objects (uniformly and without overlap), and are respectively composed of 4000, 1000 and 1000 elements. Thus, the agents only have access to a small fraction ($< 1\%$) of the environment, making the generalization problem challenging.

**Communication channel** Messages $m =: (m_j)_{j=1}^T \in \mathcal{M} =: \mathcal{V}^T$ are sequences of $T$ tokens where each token is taken from a finite vocabulary $\mathcal{V}$, finishing by a hard-coded end-of-sentence token `EoS`. In our experiments, messages have maximum length $T = 10$ and symbols are taken from a vocabulary of size $|\mathcal{V}| = 10$ to prevent a bottleneck in the communication channel.

**Speaker model** The speaker follows a recurrent policy: given an input object $x$, it samples for all $t \in [1, T]$ a token $m_t$ with probability $\pi_\theta(m_t | m_{<t}, x)$. The speaker takes in the object $x$ as a vector of size $K \times |\mathcal{X}|$ and passes it through a linear layer of size 128 to obtain an object embedding, used to initialize a LSTM [35] of size 128 with layer normalization [2]. At each time step, the LSTM's output is fed into a linear layer of size $|\mathcal{V}|$, followed by a softmax, to produce $\pi_\theta(m_t | m_{<t}, x)$

**Listener model** Given a message $m = (m_1, ..., m_T)$, the listener outputs for each attribute $k$ a probability distribution over the $|\mathcal{X}_k|$ values: $\rho_\phi^k(\cdot | m)$. The probability of reconstructing the entire object $x$ given $m$ is then $\rho_\phi(x|m) := \prod_k \rho_\phi^k(x^k|m)$. The listener passes each message $m_t$ through an embedding layer of dimension 128 followed by a LSTM with layernorm of size 128. The final recurrent state $h_T^l$ is passed through $K$ linear projections of size $|\mathcal{X}|$, each followed by a softmax, providing $K$ independent probability distributions of sizes $|\mathcal{X}|$ to predict each attribute of $x$.

**Optimization** The agents are optimized using `Adam` [42] with a learning rate of $5 \cdot 10^{-4}$, $\beta_1 = 0.9$ and $\beta_2 = 0.999$ and a batch size of 1024. For the speaker we use policy gradient [76], with a baseline

computed as the average reward within the minibatch, and an entropy regularization of $0.01$ to the speaker's loss [82]. In all experiments, we select the best models by early stopping.

## 4.2 Evaluating emergent languages properties

**Generalization** We measure generalization by computing the average test reconstruction score over all the attributes of a probe listener trained on the training set using an early stopping criterion on the validation set. Indeed, the trained listener $\rho_\phi$ may overfit to the training set, and so using it may under-estimate. Using a separate listener removes this bias.

**Compositionality** Compositionality is a fundamental feature of natural language often seen as a precondition to generalize [6, 77, 79]. We assess the compositionality by computing the topographic similarity [8, 53]. It is defined as the Spearman correlation [47, 80] between the distance in input space, i.e. the average number of common attributes, and the distance in message space, i.e. the edit-distance between the corresponding messages [54]. As we here deal with large object space and stochastic policies, we use a bootstrapped estimate of topographic similarity as in [46] to get reliable numbers. We sub-sample 1000 elements $x$ from the object space $\mathcal{X}$, and sample the corresponding message $m$ from the speaker's policy $\pi_\theta(\cdot|x)$. We compute the topographic similarity for this batch of 1000 pairs $(x, m)$. We repeat this protocol 100 times and take the mean to measure compositionality.

# 5 Empirical results

## 5.1 Visualizing the loss decomposition dynamics

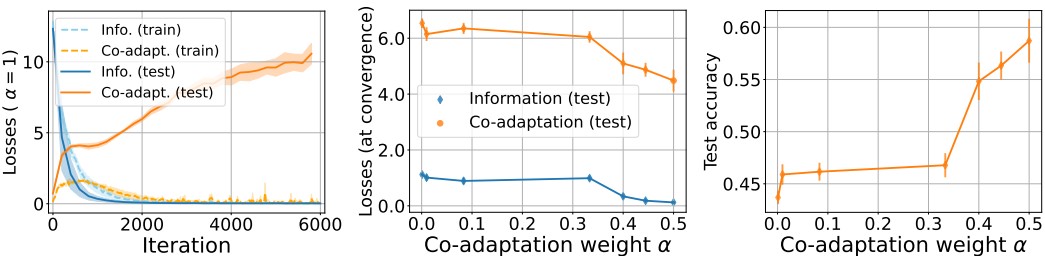

Figure 2: (a)Training dynamics ($\alpha = 0.5$). (b,c)Agents score as a function of co-adaptation weight $\alpha$.

We here visualize the loss decomposition dynamics. Following the protocol of 3.2, we control $\mathcal{L}_{\text{adapt}}^{\text{train}}$ in speaker's loss with weight $\alpha$ to understand the influence of the co-adaptation term on the language.

**The co-adaptation task overfits rapidly** We plot information and co-adaptation training dynamics in the standard setting ($\alpha = 0.5$). Note that both train and test information losses quickly converge to 0, in other words the speaker succeeds in developing a protocol that is unambiguous on both the training set and the overall distribution. On the other hand, the test co-adaptation loss diverges while the train co-adaptation keeps disminishing, highlighting a clear overfitting problem.

**The co-adaptation task promotes generalization** We then display in Figure 2 the evolution of the information and co-adaptation losses for different co-adaptation weight $\alpha$. We observe that down-weighting $\mathcal{L}_{\text{adapt}}^{\text{train}}$ tends to enforce both information and co-adaptation overfitting. Thus, even though the co-adaptation loss is not inherently necessary for the speaker to develop an unambiguous language, it is important to encourage the speaker to build a better language. This is confirmed when looking at generalization accuracies. From $\alpha = 0$ to $\alpha = 0.5$, there is a gain of $15$ points of generalization. In conclusion, we note that (i) balancing the loss in favor of $\mathcal{L}_{\text{info}}^{\text{train}}$ has a negative impact on generalization, (ii) the co-adaptation loss $\mathcal{L}_{\text{adapt}}^{\text{train}}$ pushes the speaker to develop a language that generalizes better.

These experiments highlight two key findings: (i) co-adaptation is crucial for generalization ; (ii) in standard settings, the co-adaptation loss overfits substantially, whereas the information loss does not.

## 5.2 Countering co-adaptation overfitting

We here investigate whether limiting overfitting in the co-adaptation loss may push towards languages that generalize better and are more stuctured. As described in 3.3, we compare three control baselines: *Continuous listener*, *Partial listener* with varying levels of convergence, and *Early stopping listener*.

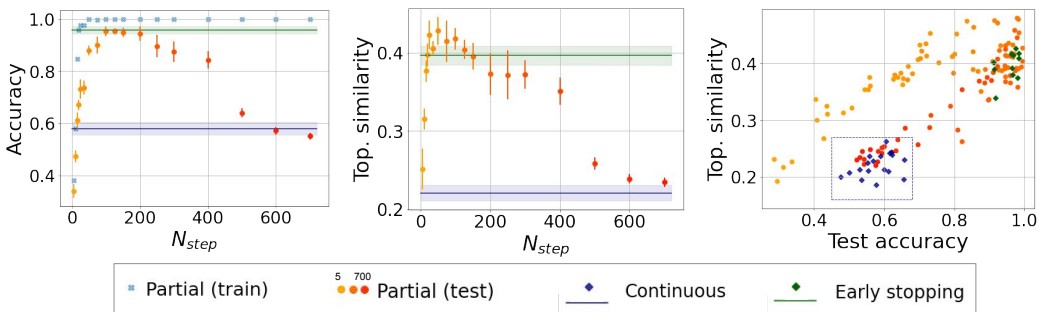

Figure 3: (a,b) Evolution of generalization and top.sim with *Partial listener*'s number of learning steps $N_{step}$ ; (c) Top. sim VS. generalization. The color level of orange dots increases with $N_{step}$. Blue (resp. green) lines and points refer to the Continuous listener (resp. Early stopping listener).

**Countering co-adaptation overfitting improves generalization** In Figure 3, we observe that the level of convergence of the *Partial listener* between each speaker's update (controlled by $N_{step}$) has a strong impact on the generalization of the emergent protocol. Overall, we recover classic machine learning trends when varying $N_{step}$: when $N_{step} < 50$, both train and test accuracy are low — the agents underfit. When $50 < N_{step} < 250$, the train and test accuracy are almost optimal — the agents are in good training regime. Finally, when $N_{step} > 250$, the train accuracy is maximal while the test accuracy collapses — the agents overfit. These observations reveal that the level of convergence of the listener has a substantial impact on the final emergent language capacity to generalize. Recall that, in these experiments, the direct effect of the listener's overfitting is mitigated, as we measure generalization using an auxiliary listener that is early stopped, and should therefore not overfit as noted in Section 4. The listener's overfitting impacts the speaker's update through the co-adaptation loss, which, by inducing a poorer final language leads to a degradation in generalization. Additionally, Figure 3 shows that the continuous listener, standard in the Lewis games literature, provides generalization performance similar to the worst overfitting listeners.

Controlling the listener's co-adaptation level appears crucial to let the speaker develop a language that generalizes well; this effect may have been underestimated in the standard Lewis learning dynamic.

**Countering co-adaptation overfitting improves compositionality** Figure 3 reveals that compositionality follows the same pattern. In the underfitting regime, the topographic similarity is low but still outperforms the *Continuous listener*. Similarly, it is also low in the overfitting regime. In-between the two — which corresponds to high generalization in Figure 3 — the topographic similarity reaches high values, which suggests that more compositional languages emerge. This indicates that the listener's lack of co-adaptation overfitting promotes structured languages.

**Compositionality correlates with generalization** In Figure 3, we plot the correlation between generalization and compositionality. As opposed to [10], we observe a strong correlation between generalization and topographic similarity when varying the *Partial listener*'s level of convergence. In particular, we identify two correlation branches: one belonging to the underfitting regime and the second to the overfitting regime. Together, they retrace the evolution of generalization and compositionality with respect to $N_{step}$. We see that *Continuous listeners* belong to the end of this trajectory, in the overfitting regime. Note that the blue rectangle — which delineates the range of values reached with the *Continuous listener* — corresponds to the classic learning setting in the literature. As this range is tight, it may explain the initial negative results reported by [10].

In conclusion, the listener exerts a necessary pressure on the speaker to develop a structured language that generalizes better. This pressure can be controlled by limiting the listener's level of overfitting, which is inevitably too high when the listener is trained continuously as is usually done.

**Comparison with standard regularization methods** In practice, re-initializing the listener as done with the *Partial* or *Early stopping listener* is costly. We thus test whether performances comparable to Figure 3 can be obtained by controlling the listener's level of overfitting with standard regularization methods. In Table 1, we report the influence of applying common regularization methods to the listener on various metrics of the language. We find that regularization consistently results in noticeable improvements. Moreover, once again, gains of generalization correlate with gains of

| | Gen. ↑ | Compo. ↑ | $\hat{\mathcal{L}}^{\text{test}}_{\text{adapt}}$ ↓ | | | Generalization ↑ | | |
|---|---|---|---|---|---|---|---|---|
| Continuous | $0.58_{\pm 0.05}$ | $0.22_{\pm 0.02}$ | $4.64_{\pm 1.22}$ | | **CelebA** | 1/1 | 1/20 | 1/100 |
| Dropout | $0.64_{\pm 0.03}$ | $0.24_{\pm 0.01}$ | $4.86_{\pm 0.52}$ | | Continuous | $0.94_{\pm 0.01}$ | $0.67_{\pm 0.02}$ | $0.39_{\pm 0.07}$ |
| No LN. | $0.70_{\pm 0.03}$ | $0.24_{\pm 0.02}$ | $4.68_{\pm 0.38}$ | | Early stopping | $\mathbf{0.97}_{\pm \mathbf{0.01}}$ | $\mathbf{0.80}_{\pm \mathbf{0.03}}$ | $\mathbf{0.69}_{\pm \mathbf{0.04}}$ |
| Weight decay | $0.72_{\pm 0.03}$ | $0.25_{\pm 0.03}$ | $4.29_{\pm 0.56}$ | | | | | |
| No LN. + WD | $0.87_{\pm 0.07}$ | $0.30_{\pm 0.03}$ | $2.12_{\pm 0.67}$ | | **ImageNet** | 1/1 | 1/20 | 1/100 |
| Early stopping | $\mathbf{0.95}_{\pm \mathbf{0.04}}$ | $0.39_{\pm 0.04}$ | $1.10_{\pm 0.69}$ | | Continuous | $0.96_{\pm 0.01}$ | $0.77_{\pm 0.01}$ | $0.51_{\pm 0.03}$ |
| Top Partial | $\mathbf{0.95}_{\pm \mathbf{0.03}}$ | $\mathbf{0.42}_{\pm \mathbf{0.02}}$ | $\mathbf{0.97}_{\pm \mathbf{0.55}}$ | | Early stopping | $\mathbf{0.98}_{\pm \mathbf{0.01}}$ | $\mathbf{0.81}_{\pm \mathbf{0.01}}$ | $\mathbf{0.64}_{\pm \mathbf{0.01}}$ |

Table 1: (left) Performance comparisons between Continuous listener, Partial listener, Early stopping listener and classic listener regularization, e.g. weight decay [34, 49], Dropout [74] and layernorm [2]. Regularization parameters were tuned and are detailed in Appendix C.1 ; (right) Generalization scores for continuous baselines and Early stopping listener on visual Lewis Games. 1/1, 1/20 and 1/100 refer to the subset ratios of the dataset.

compositionality. These trends corroborate our hypothesis that controlling the listener's learning is key to encourage the speaker to develop more structured languages. However, those methods remain under the upper bound reached by the *Early stopping listener*, which suggests that further research on regularization in cooperative games is warranted.

We complement this analysis in Appendix C.2 by studying the impact of regularization on the speaker's side, and show that such regularization does not result in similar improvements. This indicates that the listener is the main contributor to the co-adaptation overfitting.

### 5.3 Scaling to the Image Discrimination Games

To validate our empirical findings beyond synthetic games, we scale our approach to complex games with natural images as advocated by [12]. We thus train our agents on a discriminative game on top of the CelebA [57] and ImageNet [69, 19] datasets while applying previous protocol. We work on 3 sizes of training set with increasing generalization difficulty. We provide all the training details and game settings in Appendix D.1 and report our results in Table 1. While agents generalize well when trained on the entire training set, generalization issues occur on smaller training sets and performances can indeed be improved by controlling the listener's level of convergence. However, Appendix D.2 shows that gain of generalization does not correlate with gain of topographic similarity, supporting that agents' language structure is not captured by the topographic similarity in image based settings [12, 1].

## 6 Related work

The decomposition of the loss function in the Lewis Game that we introduced finds echos in the cognitive science literature. According to Skyrms [70], communicative organisms or systems are confronted with two types of information: about the environmental states shared by the agents (called *objective* information), and about how an agent would react to a signal (called *subjective* information). Communication protocols emerge as a trade-off between constraints related to those two types of information [44, 45]: the sender should be expressive [25, 24] and transcribe the information available in the world with as little ambiguity as possible, which has been described as a *bias against ambiguity* [73] ; sender and receiver should agree on the same referring system, which has been described as a *conceptual pact* [7]. The latter has been shown to impose compressibility and learnability pressures promoting structure [78, 72, 84]. This analysis resonates well with our analytical decomposition of the loss function in the Lewis game.

The first term of the decomposition, which we called the information loss, has been addressed by previous work that assumed that linguistic structure and generalization emerge from the requirement of creating an unambiguous language. In this line of work, studies have either manipulated the complexity of the environment [12, 30, 71, 60], restricted the bandwidth of the communication channel [48, 66], or added noise to the message [50, 85]. In our main experiment, we do not apply such information constraints to better focus on the second term of the decomposition, the *co-adaptation* constraint, less studied within a machine learning approach. Previous work have assumed that the co-adaptive dynamics encourage speakers to develop a more structured language for

learnability reasons [56]. Support for this hypothesis can be found directly via the implementation of a neural variant of Iterated Learning [65] or the introduction of learning speed heterogeneities [68] and indirectly via the restriction of agents capacity [66], the variation of the communication-graph in populations [28, 41] or the addition of newborn agents [15]. In our paper, we demonstrate that a co-adaptation term is always present in standard agents optimization protocols and show that controlling *co-adaptation overfitting* enhances language properties. The existence of an overfitting regime found under the default setting (continuous training) may explain the counter-intuitive lack of relationship between compositionality and generalization previously reported with neural agents [51, 10, 38, 20].

# 7   Conclusion

In this paper, we propose a methodological approach to better understand the dynamics in Lewis signaling games for language emergence. It allows us to surface two components of the training: (i) an information loss, (ii) a co-adaptation loss. We shed light that the agents tend to overfit this co-adaptation term during training, which hinders the learning dynamic and degrades the resulting language. As soon as this overfitting is controlled, agents develop compositional languages that better generalize. Remarkably, this emergent compositionality does not result from environmental factors, e.g. communication bottleneck [43], under-parametrization [48, 26], population dynamics [12, 68], memory restriction [15, 16] or inductive biases [67], but only through a trial-and-error process. Therefore, we advocate for a better comprehension of the optimization and machine learning issues. As illustrated in this paper, such understanding may unveil contradictions between computational models and language empirical observations and better expose the existing synergies between learning dynamics and environmental factors [27, 83, 64, 14, 18, 22].

## Acknowledgments

Authors would like to thank Rahma Chaabouni, Marco Baroni, Paul Smolensky, Bilal Piot and Karl Tuyls for helpful discussions and the anonymous reviewers to their relevant comments. M.R would also like to thank Michael Sander and Maureen de Seyssel for last minute feedbacks. M.R. was supported by the MSR-Inria joint lab and granted access to the HPC resources of IDRIS under the allocation 2021-AD011012278 made by GENCI. P.M. was supported by the ENS-CFM Data Science Chair. E.D. was funded in his EHESS role by the European Research Council (ERC-2011-AdG-295810 BOOTPHON), the Agence Nationale pour la Recherche (ANR-17-EURE-0017 Frontcog, ANR-10-IDEX0001-02 PSL*, ANR-19-P3IA-0001 PRAIRIE 3IA Institute) and grants from CIFAR (Learning in Machines and Brains) and Meta AI Research (Research Grant).

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
