# OpenReview forum: "Emergent Communication: Generalization and Overfitting in Lewis Games"
_NeurIPS.cc/2022/Conference — NeurIPS 2022 Accept_

### Official Review · Reviewer_mSoN · 2022-07-09

**Rating:** 8
**Confidence:** 4
**Soundness:** 4 excellent
**Presentation:** 3 good
**Contribution:** 4 excellent

**Summary:**

The paper provides a theoretical analysis of the objective of Lewis communication games, that reveal that the objective can be decomposed into two components: (1) mutual-information component, that encourages the speaker to generate unambiguous (highly informative) messages on its input; and (2) adaptation component, that encourages the distribution of the speaker's posterior to match the distribution of the listener. The latter is directly optimized by both players, and is zeroed when the listener achieves an optimal distribution over objects given a message. Following this theoretical analysis, the authors hypothesize that overfitting issues in the adaptation game result in some previously-reported problems in the emergent communication, such as low accuracy and lack of compositionality.

To test this hypothesis, the co adaption term is estimated with a probe -- a listener that is trained to convergence over the frozen speaker. By modifying the weight of the probe's loss and the original loss in the reward of the speaker, the influence of mutual information component vs. the adaption component is tested. it is shown that (1) the adaptation term severely overfits, while the information term does not (2) by replacing the continuous co-training of the speaker and the listener with an alternate training where the listener is trained for N steps, it is possible to find a setting where the emergent language presents much superior properties.

**Questions:**

- is it accurate to say that the proposed weighting scheme can be equivalently describes as follows? (1) we know the overall value of the loss L, and we can estimate the information term L_info (which doesn't require marginalization) with a probe; (2) Then the adaptation term is approximated as L_adapt = L - L_info; (3) Instead of training with L, we train with alpha*L_adapt + (1-alpha) * L_info.  In case this description is correct, I think it's a more intuitive way to describe the weighting experiment.

- Why does the adaptation term strongly overfit, while the other term does not?

- It would be beneficial to briefly explain why the speaker's posterior is the optimal distribution over objects for the listener. Intuitively, why can't the listener make up for some ambiguity in the speaker's messages by changing its distribution?

**Limitations:**

Yes.

**Strengths And Weaknesses:**

Strengths: the paper is an excellent example of a simple and highly elegant theoretical observation that leads to significant empirical gains. It is largely clearly written (apart from some issues, see below), and the empirical results (especially figure 2 and 3) are persuasive and surprising -- they point out to an inherent limitation that arises from optimization and may explain many of the previously reported issues with neural Lewis games. As such, the analysis in this paper has the potential of substantially influencing the study of emergent communication, and I strongly support its acceptance.

Weaknesses: while the presentation is largely clear, it would be beneficial to improve, in the camera ready version, the presentation of the experiment that involves the weighting of the probe's loss and the original loss. In equations 10-11 the constant alpha is presented as a way to *upweight* the original loss, while, in practice, it is used to *downweight* the influence of the adaptation term. While the two are equivalent, these two conflicting views make it harder to understand the discussion in lines 237-244 (I think this paragraph in particular has to be improved. particularly the sentence at line 240).

---

> ### Author Response · Authors · 2022-08-01
> **Response to Reviewer mSoN**
>
> We thank the reviewer for their insightful comments.
>
> **1/** The reviewer's main concern is the lack of clarity of the weighting experiment. This remark is related to this first question asked by the reviewer:
>
> > *Is it accurate to say that the proposed weighting scheme can be equivalently describes as follows? (1) we know the overall value of the loss L, and we can estimate the information term L_info (which doesn't require marginalization) with a probe; (2) Then the adaptation term is approximated as L_adapt = L - L_info; (3) Instead of training with L, we train with alpha*L_adapt + (1-alpha)*L_info. In case this description is correct, I think it's a more intuitive way to describe the weighting experiment.*
>
> This is indeed an accurate description. We thank the reviewer for this suggestion, which we aim to incorporate in the paper as it is more intuitive. Note that there is a caveat to this interpretation: α can only be taken in the range [0,0.5] as explained in Appendix B.
>
> In short, the probe listener only provides a (potentially tight) upper bound estimate \hat{L_info}  of  L_info. Replacing L_info by \hat{L_info} is only sensible when we aim at minimizing L_info, i.e. if we keep a positive coefficient in front of L_info in the loss. In practice, if we were to train with α*\hat{L_adapt} + (1-α) * \hat{L_info}, we would actually train α*L_listener + (1-2α)*L_probe. In the range alpha=0.5 to alpha=1, the term (1-2α) becomes negative and we have no guarantee that the system actually minimizes L_info by minimizing L_probe=\hat{L_info}.
>
> However, it is true that the formulation L = α L_adapt + (1-α) L_info is more intuitive. We will update the protocol, plot and discussion in the final version using this formulation and mention the reason why α can only be taken in the range [0,0.5].
>
> **2/**
>
> > *Why does the adaptation term strongly overfit, while the other term does not?*
>
> The reviewer raises an interesting question. Unfortunately, we could not come up with experimental results to rigorously claim causal reasons showing why the co-adaptation overfits while the information loss does not. Explaining this phenomenon constitutes an interesting avenue for future work.
>
> Having said that, this is  our thought on this point:
>
> The speaker and listener have two different optimization processes. The speaker is trained by reinforcement while the listener is optimized in a supervised fashion. Therefore, the speaker could be naturally regularized by the variance of its reinforcement objective (limiting the information loss overfitting) while the listener’s supervised objective has lower variance leading to stronger overfitting.
>
> **3/**
>
> > *It would be beneficial to briefly explain why the speaker's posterior is the optimal distribution over objects for the listener. Intuitively, why can't the listener make up for some ambiguity in the speaker's messages by changing its distribution?*
>
> The reviewer raises a good point. The speaker’s posterior is optimal because we are using a cross entropy as objective. For other objectives, we might end up with a more or less picky distribution over objects. The derivation in the cross-entropy case is provided in Appendix A.1.1. and we added a pointer to this appendix in the revised version (Note: the general case is described in Appendix A.1.2). Do you think it would make sense to put the explanation in the main paper ?

---

> > ### Comment · Reviewer_mSoN · 2022-08-09
> > **Response**
> >
> > I appreciate the authors response, which adequately addressed my questions, and I keep my "strong accept" assessment.

---

### Official Review · Reviewer_VUeC · 2022-07-11

**Rating:** 8
**Confidence:** 3
**Soundness:** 4 excellent
**Presentation:** 4 excellent
**Contribution:** 3 good

**Summary:**

The authors propose a framework for studying the generalization and compositionality of communication games playing Lewis signaling games. In short, there are two agents a speaker and a listener and "the speaker observes a random state from its environment, e.g. an image, and sends a signal to the listener. The listener then undertakes an action based on this signal. Finally, both agents are equally rewarded based on the outcome of the listener’s action." (Line 24-26) The authors demonstrate how the agent's loss can be decomposed into an information term that quantifies the degree of ambiguity of the language protocol and a co-adaptation term that quantifies the gap between the listener and its optimum.

This observation allows studying the source of overfitting (or lack of generalization) in the communication agents especially when communication is in an unseen state from the training set. The main idea is to decrease the importance of the co-adaptation term in the loss so as to avoid the speaker to overfit the listener. For doing so, the authors experimented with using different training strategies Continuous listener, Partial listener, and Early stopping listener (Section 3.3).

The authors experimented using, the EGG toolkit with an environment in which every object has K = 6 attributes, each taking 10 different values, for a total of 1 million objects, and a non-overlapping train/val/test split is used. The experiments confirm the hypothesis of the loss decomposition by showing the importance of the co-adaptation term while evaluating an unseen set of objects (from the test set). Moreover, the authors show the effectiveness of the proposed method on more complex games CelebA and ImageNet, showing a much stronger generalization compared to a "standard" Continuous listener.

**Questions:**

I have no particular question. The paper is very clear and self-explanatory, I appreciated the read.

**Limitations:**

Not reported by the authors.

**Strengths And Weaknesses:**

Strengths
- The proposed loss decomposition is novel, to the best of my knowledge. The paper is formal and rigorous without being too verbose or hard to read in its presentation.
- The experimental results are conclusive and well-conducted. Adding the final experiments on more complex tasks adds validation to the proposed hypothesis.
- The paper is very well written and easy to follow. The content is well organized and the plots add to the presentation.

---

> ### Author Response · Authors · 2022-08-01
> **Response to Reviewer VUeC**
>
> We would like to thank the reviewer for their encouraging comments and we are glad they enjoyed reading the paper. If you have any new comments, please do not hesitate to reach out in the discussion.

---

### Official Review · Reviewer_dJK1 · 2022-07-13

**Rating:** 5
**Confidence:** 4
**Soundness:** 4 excellent
**Presentation:** 4 excellent
**Contribution:** 2 fair

**Summary:**

Teaching agents to play lewis signalling games is still one of the most common ways of studying emergent communication, and has historically been a test bed for our understanding of many EC dynamics.  In this paper, the authors propose to view the standard lewis game loss as a composition of two terms -- one that essentially promotes a unique message for each object, and one that promotes mutual understandability between speaker and listener.  It is this second term that is the main focus here, deemed as co-adaptation, as the authors explore the effect that artificially controlling this factor has on the learned languages.  They find that when trained in the de facto way, this term often overfits, and this overfitting has a detrimental effect on compositionality.


**Questions:**


I think the authors did a convincing job at showing how this co-adaptability effect gets to the heart of effective communication protocols failing to exhibit compositional structure and generalization, so as a take-away I think that is constant, but I'm unsure from a practical perspective the extent to which it matters to regularize the term explicitly vs. obtain regularization from aspects of the game/environment.

To this end, to what extent do these results persist as the task changes?  I was a bit disappointed that in scaling it up to the vision task, the size of the task was deliberately kept low.  Is it difficult to run on the full set?  Could the authors present some results more in-line with what other EC work is focusing on?  Perhaps it is not important to consider ad hoc regularization of the loss function if the environment is sufficiently rich or the task sufficiently hard.

Albeit the decomposition and analysis are novel, but can the authors clarify how their focus on the co-adaptability factor and early stopping regularization is different from the goal and solution of neural iterated learning and ease of teaching (both cited)?  It seems that these papers have identified this dynamic as a point where compositionality and generalization fail.

**Limitations:**

In terms of limitations, not sufficiently.  In terms of societal impact, it should not be applicable.

**Strengths And Weaknesses:**

Strengths:

First off, many thanks to the authors for presenting such a clear paper.  It was generally a joy to read.  The proposed decomposition makes a lot of sense, and maybe more than that just formalizing the loss in this way may help other researchers to structure the hypotheses of their own work.  In particular I liked the formalization of l^train_adapt and its role in the loss function since this gets quite close I think to how one should discuss generalization to novel object-attribute combinations through compositionality.

In terms of the results, the authors make a compelling case for the relationship between co-adaptation overfitting hindering compositionality in languages.  Both experiments with really hands-on early stopping regularization, and with more traditional ML regularization on the co-adaptation term, yield better generalization.


Weaknesses:

Although I think this paper is mostly ready for publication in its current state, and I do think it is beneficial to the EC community, I'm not sure the scope of the contribution is significant enough to warrant publication at NeurIPS, both in terms of what benefit it brings to the EC community, and in terms of it's interest to the larger set of research at the venue.  I think the contribution is quite niche (okay) and maybe rehashing a lot of what the EC community seems to know.

Does this work really shine new light on these issues, or does it just reinforce the lessons learned by the slew of other EC work?  I think it comes down to this: other (cited) EC work tends to also tackle the same problem of lack of compositionality stemming from co-adaptation, but focus more on how solutions may emerge from population dynamics and which sets of agents are communicating with each other.  These will ultimately find themselves expressed in gradients.  In this work, the authors cut straight through to the optimization itself.  In that sense, it helps point research towards more desirable properties in the loss function, but it is still up to the research community to find plausible ways for these pressures to originate.  Because presumably as a model of language development we don't want strong assumptions governing the loss itself over the actual agent/environment dynamics?  And the particular method of regularizing as applied here (primarily the early stopping) seems hacky and more for the purpose of probing than for simulation, whereas other proposed methods like neural iterated learning apply similar constraints in a more plausible way.

So in that extent, the closing argument (L342) seems somewhat backwards to me -- researchers seem aware that co-adaptation and lack of compositionality is undesirable, so they design realistic and intuitive pressures which ultimately have similar effects on the loss.  It is not surprising if a more hands-on approach goes directly to the loss to accomplish the same effect - as the authors say, "remarkably, this emergent compositionality does not result from [things]..." but it is instead created forcefully, and not in some natural way that would be remarkable IMO.

In conclusion, like a good literature review, I would like to see it published - it could be useful to EC researchers.  But I find it difficult to see this particular analysis as sufficiently novel or important to future EC research.

---

> ### Author Response · Authors · 2022-08-01
> **Response to Reviewer dJK1 (1/2)**
>
> First, we would like to thank the reviewer for their review and insightful remarks.
>
> The reviewer’s main concern is about the contribution of the paper. We would like to answer this concern in three points:
>
> **1/ The goal, method and conclusion of our experiments are different from Neural Iterated Learning and Ease-of-teaching.** The two closest papers to the empirical part of our paper are Ease-of-teaching [1] and iterated learning based approach such as Neural Iterated Learning [2]. However, we differ from those works in goal, method, and conclusions.
>
> *Goal* : Ease-of-Teaching and NIL propose techniques inspired by human language evolution to strongly regularize the system (as we mentioned in the Related Section). Our experiments aim at understanding the cause of the emergence/non-emergence of compositionality/generalization rather than proposing solutions to get compositional emergent languages that generalize well. In other words, our work focuses on diagnosing the root cause of a problem, while Ease-of-teaching and NIL propose efficient and plausible ways to cope with this co-adaptation overfitting problem.
>
> *Method* : NIL introduces the concepts of generation: the agents are iteratively pre-trained by imitating the previous generation, then fine-tuned through interaction. In EoT, both the speaker and the listener are trained by interaction, but the listener is reset on a regular basis. There is no imitation phase nor speaker-listener generation per se. In our work, we train the speaker by interaction, but at every timestep, we train a new listener from scratch for a given number of steps by freezing the speaker, before applying a single gradient step to the speaker. As opposed to NIL, there is no imitation phase nor speaker-listener generation. As opposed to EoT, the listener is never trained by interaction (it is only trained with the frozen speaker to control the level of training). As mentioned in the paper, this training regime is computationally intense, but our goal is to diagnose an optimization problem.
>
>
> *Conclusion* : While NIL and ease-of-teaching provide plausible solutions to enable emergence of compositionality/generalization, we provide a machine learning focused explanation for why regular emergent communication  setups fail in displaying such property in the first place. In our paper, resetting is used to demonstrate that the degree of learning of the listener has a huge impact both on compositionality and generalization.  It allows us to unify the problem of compositionality/generalization under a single ML concept, namely co-adaptation overfitting, and provide clues to interpret empirical findings.
>
> **2/ The theoretical analysis is novel.** As far as we know, the main contribution of the paper, which is the theoretical part and the loss decomposition, is novel. It aims at understanding the learning components of the game and may inspire empirical works. Moreover, the methodology per se is a contribution as intuition can be derived from those techniques, especially since those tools can scale to populations.
>
> **3/ Reviewers agree on the novelty of the paper.** As the reviewer mentioned, the loss decomposition is novel and could help other researchers structure the hypotheses of their own work. Both reviewers mSoN and VUeC also agree that the theoretical contribution is novel and could be of great help for the EC community. Eventually, reviewer mSoN notes that not only the theoretical part is significant but also empirical results are *“surprising”* and that the *“analysis in this paper has the potential of substantially influencing the study of emergent communication”.*
>
> Overall, we believe our contribution to be substantial for the EC community because it helps understanding the limitations of the current computational models and removes potential confounding factors in the simulation of language emergence.
>
> **References**
>
> *[1] F. Li and M. Bowling. Ease-of-teaching and language structure from emergent communication. In Proc. of Advances in Neural Information Processing Systems (NeurIPS), 2019.*
>
> *[2] Y. Ren, S. Guo, M. Labeau, S. B. Cohen, and S. Kirby. Compositional languages emerge in a neural iterated learning model. In Proc. of International Conference on Learning Representations (ICLR), 2020.*

---

> > ### Author Response · Authors · 2022-08-01
> > **Response to Reviewer dJK1 (2/2)**
> >
> > We now try to answer direct questions asked by the reviewer.
> >
> > **1/**
> >
> > > *I think the authors did a convincing job at showing how this co-adaptability effect gets to the heart of effective communication protocols failing to exhibit compositional structure and generalization, so as a take-away I think that is constant, but I'm unsure from a practical perspective the extent to which it matters to regularize the term explicitly vs. obtain regularization from aspects of the game/environment.*
> >
> > One conclusion of our work is to relate compositionality and generalization under a unique optimization phenomenon: co-adaptation overfitting. Then, numerous techniques can be used to regularize the system. In the paper, we use early stopping and standard regularization techniques to shed light on the phenomenon. Regularizing by the complexity of the task/environment or even with other architectures would also be alternative solutions. In the paper, we do not intend to debate which source of regularization is more plausible in explaining human language emergence (which is actually a long standing debate in cognitive science and language evolution).
> >
> > > *To this end, to what extent do these results persist as the task changes? I was a bit disappointed that in scaling it up to the vision task, the size of the task was deliberately kept low. Is it difficult to run on the full set? Could the authors present some results more in-line with what other EC work is focusing on? Perhaps it is not important to consider ad hoc regularization of the loss function if the environment is sufficiently rich or the task sufficiently hard.*
> >
> > For experiments with images, we added the results on the entire dataset as recommended and wrote a short comment (in red) in the revised paper.
> > In those experiments, agents reach almost perfect test accuracy even with the standard setting. We suspect that the score is explained not only by the richness of the environment but also by the simplicity of the discrimination task.
> > In the context of our paper, we try to understand the positive/negative results displayed in past works about compositionality/generalization. That is why we first reported smaller dataset sizes for which generalization is more challenging.
> >
> > *Note*: we added the scores on the entire dataset in the main paper. Results on generalization and topographic similarity for all the other types of regularization must still be added into the appendix, and we will add them after the rebuttal due to computation time constraints.
> >
> > **2/**
> >
> > > *Albeit the decomposition and analysis are novel, but can the authors clarify how their focus on the co-adaptability factor and early stopping regularization is different from the goal and solution of neural iterated learning and ease of teaching (both cited)? It seems that these papers have identified this dynamic as a point where compositionality and generalization fail.*
> >
> > Please see the Response (1/2). We are happy to discuss this point further.

---

> > > ### Comment · Reviewer_dJK1 · 2022-08-09
> > > **Looking for more practical/fundamental significance to EC work**
> > >
> > > Thanks to the authors and reviewers for the additional discussion points.  I think in some sense this is a non-answer, as much as my original review was a non-criticism: this is a solid paper and deserves to be accepted even in its current state, at some venue.  But in terms of the scope, significance, and general relevance to the NeurIPS community, it seems a bit lacking relative to the extremely high bar of other areas.  I agree with essentially all the points of the rebuttal, but that discussion does not seem to further my opinion of the paper in these areas.
> > >
> > > I appreciate the authors presenting the additional experiments scaling the experiments up to the full image data, as this is the type of additional exploration I alluded to earlier.  However, the result is somewhat in line with my intuition -- that these regularization methods seem important in extremely small scenarios, which (albeit completely common in this research area) are very poor approximations of the actual scale, in almost every dimension, in which language evolved.  As the environments for this research are created almost uniquely for each research project,  and because the environment/task has such a dominating effect on how the communication protocols are learned, it's also not clear to what extent these regularization methods are important for controlling across a variety of environments.  If the experiments instilled confidence that many typical current EC research would benefit from such regularization, that typical and future EC experiments would not completely subsume these differences instead through the richness or scale of the experimental/task, it would be easier to argue for the lasting significance of the work.
> > >
> > > I believe there is value in this decomposition, even if it is largely a conceptual one.  But I'm not convinced that it's necessarily important in a practical sense against modest increases in the scale of the experiments.  I would have liked to have seen more evidence to the pervasive importance of these factors, that the performance gaps do not narrow easily as other variables in the experiments are changed, to warrant a stronger recommendation.

---

### Author Response · Authors · 2022-08-01
**General response**

We first want to thank the reviewers for the high quality of their response and their insightful feedback. We hope to have correctly answered their questions, and we are happy to continue the discussion to keep improving the paper.

We have uploaded a revised version of the paper to include the insightful remarks of the reviewers:

- We added the results of the visual discrimination game on the entire image datasets with short comments (Table 1 and Section 5.3)

- We added a reference to Appendix A.1.1 in Section 2.2 (line 90) to point toward the detailed explanation of the expression of the Optimal listener

- We will update the protocol, plot and discussion of the weighting experiment after the rebuttal using the formulation proposed by Reviewer mSoN (see Response to Reviewer mSoN for additional details). The results are unchanged with this formulation.

---

### Meta-Review · Area_Chair_anpe · 2022-08-21

**Recommendation:** Accept
**Confidence:** Certain

**Metareview:**

This paper shows that the objective for Lewis games, as treated in the recent emergent communication (EC) literature, can be decomposed into two losses: an information loss (whether each message refers to a unique referent) and a co-adaptation loss (which quantifies the alignment of the speaker and listener). It shows that lack of generalization in unregularized EC is mostly due to the latter, and empirically shows that intervening on co-adaptation via regularization (early stopping/reinitialization) improves generalization.

The reviewers are generally positive about this paper. The one somewhat negative score is actually quite positive as well: the reviewer concedes that “this is a solid paper and deserves to be accepted even in its current state, at some venue”, but questions the overall impact of the contribution and whether it merits publication in NeurIPS. Unfortunately a fourth review, although promised by the reviewer, did not materialize in time, even after repeated prodding, so I had to provide it myself (see below).

As an area chair, I am somewhat torn about this paper. It is well-written and I think the field will benefit from having these things made more explicit. Sadly, I also think it shows very clearly how frustratingly little progress EC has made as a field. We are still talking about the same things, years later after its revival, in uninteresting unrealistic toy settings looking for “linguistics” and “compositionality” when basic information/communication theory combined with basic optimization would probably be more adequate. Overall, I think this paper can help the field move in the right direction and I am recommending conditional acceptance: the notation needs to be shored up, the assumptions and limitations need to be made much more explicit, and it needs to be made suitable not only for readers intricately familiar with EC, but also for readers who are only just reading their first paper on the subject.

—

More detailed AC review:

Strengths:
* Presentation: This paper is well-written, the decomposition makes a lot of sense and will not come as a surprise to anyone working in the field.
* Soundness: The experiments and evaluation are thorough and appear to be easily reproducible given the details provided. The paper provides additional experiments on different more “complex games” to overcome potential criticisms of its toy task nature.
* Impact: The EC literature, or even the field of EC broadly speaking, is extremely troubled by a poor understanding of how basic assumptions (eg questions of optimization, setup) impact observations (eg compositionality) and there is an extraordinary amount of wheel-reinvention, exacerbated by the lack of a standardized evaluation protocol. This paper has the potential to help practitioners be more explicit about their assumptions.

Weaknesses:
* Applicability and novelty: The main contribution of this paper, in my mind, is showing the decomposition and using it to elucidate the impact of training dynamics on the emerging communication protocol. However, this decomposition only applies in a limited setting and that assumption is not nearly made explicit enough. The final loss is negative log likelihood and the speaker is trained via policy gradients, without any constraints or regularization (either on the communication channel or the listener supervision). Almost all papers in EC are exactly about what constraints/regularization/dynamics we can impose in order to obtain better generalization. I think the distinction (i.e. “decomposition”) between decodability (“information”) and learnability (“adaptation”) is well-known amongst serious EC practitioners, and almost any paper published on the subject deals with these in some form.
* Clarity: Given the above observation, and the fact that the decomposition follows from trivial math, the exposition itself is valuable if it makes something very explicit that was heretofore not explicit enough, such that future work will benefit from it being explicit. The paper has the potential to do this, but in my opinion, disappointingly, falls short: too much of the writing assumes too much prior knowledge on the part of the reader. For this paper to be maximally valuable, I would want it to be understandable by someone who doesn’t know anything about emergent communication and reads this as their very first paper. This issue is particularly prevalent in the most important part, Section 2 and the corresponding appendix: the notation in the proofs is almost unforgivably convoluted; the sub- and super-script mixing for different parameterizations is unnecessarily confusing, especially with the listener parameterization \phi never actually being introduced as such; the actual proofs in the appendix never making explicit that the two losses concern the speaker and listener respectively (try having this read by someone unfamiliar with the field, they’d instantly be lost); and all of it is basically building on the work from a very specific group of people who do things in a very particular way using a framework (EGG) that nobody else uses, without ever making explicit that other EC papers do things very differently (there are definitely EC papers that do early stopping, freezing/probing in different phases of training, etc. but they tend to study what constraints can be imposed on top of the standard task formulation to make things generalize, usually in much more sophisticated games). I also did not like that the communication channel itself was not bottlenecked, with |V|=T=10 — in this setting everything collapses to basic information theory, and the assumption is not realistic for studying any emergent linguistic phenomena.

Overall, if it was up to me, I would have written this paper very differently, with an argument along the lines of: the EC literature is a mess, let’s make things more explicit by decomposing the loss in the most basic EC setting, and then we can understand all of the interventions in the prior EC literature (eg ensembles/freezing/populations/grounding/constraints/regularizations/etc) in this new light, and we can even come up with some new approaches to tackling this problem, such as down-weighting co-adaptation. I think the paper has merit, and I think it can be accepted but it really has to address its shortcomings: Section 2 and the proofs have to be notationally extremely clear with the proofs written out in much more detail and the paper has to make its assumptions much more explicit, i.e. that it applies to the limited basic setting that was mostly studied by a very limited group of people.

Typos I came across:
- “and experimental choice” - choices
- “to ease the reader intuition” - reader’s
- “deep model are large enough” - models
- “the train co-adapation keeps dismishing” - co-adaptation, diminishing

**Award:**

No

---

### Decision · Program_Chairs · 2022-09-14

Accept